# Genome-Wide Characterization of Dirigent Proteins in *Populus*: Gene Expression Variation and Expression Pattern in Response to *Marssonina brunnea* and Phytohormones

**Lingling Li, Weibo Sun, Peijun Zhou, Hui Wei, Pu Wang, Hongyan Li, Shamsur Rehman** [ID]**, Dawei Li** [ID] **and Qiang Zhuge \*** [ID]

Co-Innovation Center for Sustainable Forestry in Southern China, Key Laboratory of Forest Genetics & Biotechnology, College of Biology and the Environment, Nanjing Forestry University, Ministry of Education, Nanjing 210037, China; linglingli@njfu.edu.cn (L.L.); czswb@njfu.edu.cn (W.S.); pjzhou@njfu.edu.cn (P.Z.); HW@njfu.edu.cn (H.W.); wangpu@njfu.edu.cn (P.W.); lhy1110@njfu.edu.cn (H.L.); shamsurrehman@ahau.edu.cn (S.R.); dwli@njfu.edu.cn (D.L.)

\* Correspondence: qzhuge@njfu.edu.cn; Fax: +86-25-85428701

**Abstract:** *Marssonina brunnea* causes a major disease that limits poplar growth. Lignin and lignan play essential roles in protecting plants from various biological stresses. Dirigent (DIR) proteins are thought to control the stereoselective coupling of coniferyl alcohol in the formation of lignan and lignin. DIR family members have been well studied in several plant species, but no previous detailed genome-wide analysis has been carried out in forest trees, such as poplar. We identified 40 *PtDIR* genes in *Populus trichocarpa* and classified them into three subgroups (DIR-a, DIR-b/d, and DIR-e) based on phylogenetic analyses. These genes are distributed on 11 poplar chromosomes, and 80% of *PtDIRs* (32/40) are intronless. The *cis*-element analysis inferred that *PtDIRs* possess many types of biological and abiotic stress-response *cis*-elements. We also analyzed intra- and inter-specific collinearity, which provided deep insights into the evolutionary characteristics of the poplar *DIR* genes. Analyses of the protein tertiary structure and critical amino acid residues showed that PtDIR7–10 and PtDIR13–16, which belong to the DIR-a subfamily, might be involved in the regio- and stereo-selectivity of bimolecular phenoxy radical coupling in poplars. Quantitative reverse transcription polymerase chain reaction (RT-qPCR) analysis revealed different expression patterns for the *PtDIR* genes of *P. trichocarpa* and the *PeDIR* genes of 'Nanlin 895' in various tissues. Additionally, we analyzed responses of PeDIRs to *M. brunnea* and different phytohormone treatments (abscisic acid, salicylic acid, methyl jasmonate, and ethylene) in 'Nanlin 895'. The results showed that at least 18 genes responded strongly to *M. brunnea*, and these *PeDIRs* also showed significant responses to phytohormones. These results suggest that *DIR* genes are involved in the poplar defense response against *M. brunnea*, and this study will provide fundamental insights for future research on poplar *DIR* genes.

**Keywords:** DIR family; *Marssonina brunnea*; genome-wide analysis; gene expression; phytohormones; lignan and lignin synthesis

## 1. Introduction

Poplar is a perennial deciduous tree with a wide range of uses, such as for wood and industrial timber [1,2]. However, woody plants have a long growth cycle and are vulnerable to diseases, insect pests, and other stresses [3]. Poplar black spot, which is mainly caused by *Marssonina brunnea* is a major disease that limits poplar growth. In poplars, *M. brunnea* infection leads to premature defoliation and seriously restricts growth [4–6]. Therefore, it is essential to investigate *M. brunnea* resistance genes related to *M. brunnea*. Phytohormones, including abscisic acid (ABA) [7,8], salicylic acid (SA) [9], methyl jasmonate (MeJA) [4,6,10], ethylene (ETH) [6], play an important role in the process of poplar disease resistance.

Exploring the expression patterns of poplar genes resistance to *M. brunnea* in response to different phytohormones can help us to understand the regulatory mechanism of these genes better.

DIR proteins were first identified in *Forsythia suspensa* in 1997 [11]. Without FiDIR1, the oxidative coupling reaction of coniferyl alcohol cannot perform regio or stereoselectivity, making the product racemic. With the participation of FiDIR1, the reaction can restore both regio and stereoselectivity, and the product produced is (+) pinoresinol [11]. This indicates that DIR proteins play important roles in the correct formation of lignans. DIR proteins were subsequently reported in a variety of monocotyledon and dicotyledon plants, such as western red cedar (*Thuja plicata*) [12], *Schizandra chinensis* [13], *Arabidopsis thaliana* [14], rice (*Oryza sativa*) [15], wheat (*Triticum aestivum*) [16], soybean (*Glycine max*) [17] and cotton (*Gossypium hirsutum*) [18]. The localization sites of DIR proteins were consistent with the deposition and biosynthesis sites of lignin [19]. So, DIR proteins also play essential roles in guiding the correct formation of lignin. Lignin is an important component of the cell wall and is essential in resistance to adverse external environments and provides a physical barrier for healthy plant growth [20]. Lignan is a dimer formed by the polymerization of lignin monomers and can reduce the toxic effects of pathogens on plant cells [21]. Phylogenetic analyses of many *DIR* genes from different species have been conducted, and the whole *DIR* gene family is divided into six subfamilies: DIR-a, b/d, c, e, f, and g [22,23]. Experiments have shown that some members of DIR-a are directly involved in the stereoselective coupling reaction of lignans and lignin, while there are fewer relevant studies demonstrating that the genes of other subgroups have a directive function [23,24].

*DIR* genes can respond to many types of biotic and abiotic stresses. In terms of responding to abiotic stresses (salt, drought, high/low temperature, pesticide residue, waterlogging, and $H_2O_2$), there is evidence of *ScDIR* in sugarcane [25], *OsDIRs* and *ShDJ* in rice [26,27], *BrDIRs* in *Brassica* [28], *BhDIR1* in *Boea hygrometrica* [29], and *CsDIR16* in cucumber [30] responding to them. In terms of resistance to insect and pathogen stresses induced by weevils or mechanical damage, six *DIR* genes from Sitka spruce (*Picea sitchensis*) bark can be expressed rapidly and strongly (up to 500-fold) [22]. In soybean, the expression of *GmDIR22* was upregulated after inoculating with *Phytophthora sojae*. Compared to the wild type, soybean overexpressing GmDIR22 showed a significant increase in the total accumulation of lignans and enhanced resistance to *P. sojae* [17]. In addition, pathogens such as *Fusarium solani* f. sp. *phaseoli* of soybean [31], *Colletotrichum gloeosporioides* of *Physcomitrella patens* [32], *Erysiphe necator* of *Vitis vinifera* [33], and *Verticillium dahliae* of cotton [19] can also induce high expression of *DIR* genes in the corresponding plants.

Currently, few studies exist on the poplar *DIR* gene family, and the contribution to disease resistance in poplar of each subfamily member is not apparent. In this study, we used a bioinformatics approach to identify *PtDIRs* from the whole *P. trichocarpa* genome, and we analyzed the classification, conserved motifs, gene structures, and chromosome distribution. To explore candidate genes associated with lignin and lignan metabolism in poplar, we analyzed the protein tertiary structures of the DIR-a subfamily to predict the possible functions of the PtDIR proteins. Moreover, we analyzed tissue expression, *M. brunnea*-induced expression, and phytohormone response patterns. The current study will provide a reference and basis for the role of the *DIR* gene family in the poplar defense response against *M. brunnea* as well as future insights for research on the poplar *DIR* gene family.

## 2. Materials and Methods

### 2.1. Identification and Characterization of the DIR Proteins in Populus trichocarpa

The complete *P. trichocarpa* genome and protein sequence database were downloaded from Phytozome v12.1 (https://phytozome.jgi.doe.gov/pz/portal.html, accessed on 8 March 2020). To identify DIR candidates of *P. trichocarpa*, the hidden Markov model (HMM) file corresponding to the DIR domain (PF03018) was downloaded from the Pfam protein family database (http://pfam.sanger.ac.uk/, accessed on 8 March 2020) and HM-

MER 3.0 was used to search for *PtDIRs* in the *P. trichocarpa* genome database; the default parameters were adopted, and the cutoff value was set to 1.2e-28. After manual removal of the redundant sequences, the output putative DIR protein sequences were submitted to the National Center for Biotechnology Information Conserved Domain Database (CDD) (https://www.ncbi.nlm.nih.gov/Structure/cdd/cdd.shtml, accessed on 15 March 2020), Pfam (http://www.pfam.org/, accessed on 15 March 2020), and SMART (http://smart.embl-heidelberg.de/, accessed on 15 March 2020) to confirm the conserved DIR domain. These selected PtDIRs were named based on their positions on the chromosomes. We used the ExPasy website (http://web.expasy.org/protparam/, accessed on 5 January 2021) to obtain sequence length, molecular weight, isoelectric point (pI), and subcellular localization predictions of the identified DIR proteins. The Plant-mPLoc server (http://www.csbio.sjtu.Edu.cn/bioinf/plant-multi/, accessed on 5 January 2021) was used to predict the subcellular location. N-glycosylation sites (asparagine residues) of the *P. trichocarpa* DIR proteins were searched online using the NetNGlyc 1.0 server (http://www.cbs.dtu.dk/services/NetNGlyc/, accessed on 10 April 2020).

### 2.2. Phylogenetic Analysis, Classification, Protein Sequence Alignment, and Prediction of Protein Tertiary Structures (3D) of the DIR Gene Family

To investigate the phylogenetic relationship between *PtDIRs* and other *DIRs* from various plants species, *PtDIRs* were aligned with 27 *AtDIRs* from *A. thaliana* [13], 42 *OsDIRs* from *O. sativa* [11], *pdh1* from *G. max* [34], *FiDIR1* from *Forsythia intermedia* [11], two *TpDIRs* (*TpDIR5, 8*) from *T. plicata* [8], three *LuDIRs* (*LuDIR1, 5, 6*) from *L. usitatissimum*, two *IiDIRs* (*IiDIR1, 2*) from *Isatis indigotica* [35], two *GhDIRs* (*GhDIR3, 4*) from *G. hirsutum* [36], and seven *PDIRs* (*PDIR23-26, 28, 30, 35*) from *P. sitchensis* [23] using ClustalX version 2.0. All *DIR* gene sequences from the different species are listed in Supplemental Data S1. The phylogenetic tree was constructed using MEGA 6.0 (https://www.Megasoftware.net/history.php, accessed on 5 January 2021) using the neighbor-joining method with 1000 bootstrap replicates. ClustalX 2.0 was used for protein sequence alignment of the PtDIR-a subfamily. Protein tertiary structure prediction was performed using SWISS-Model (https://swissmodel.expasy.org/, accessed on 3 January 2021) and the Chimera software according to Kim's method [37].

### 2.3. Gene Structure, Conserved Motif Analysis, and cis-Acting Elements of DIRs in P. trichocarpa

The exon–intron organization of *PtDIRs* was determined by comparing the predicted coding sequence (CDS) with the corresponding full-length sequence using the Gene Structure Display Server (GSDS: http://gsds.cbi.pku.edu.cn, accessed on 15 May 2020). We used MEME (https://meme-suite.org/meme/tools/meme, accessed on 15 May 2020) for protein sequence analysis to identify conserved motifs in the identified PtDIR proteins. The specific setting parameters were as follows: 6 < motif width <50; 10 motifs were identified. To identify the various *cis*-acting regulatory elements in the promoters of the *DIR* genes, PlantCARE (http://bioinformatics.psb.ugent.be/webtools/plantcare/html/, accessed on 20 December 2020) was used to extract the region 1500 bp upstream of the CDS.

### 2.4. Chromosomal Location, Gene Duplication, and Calculation of Ka/Ks

The chromosomal distribution of *PtDIRs* was obtained from the *P. trichocarpa* genomic database and location mapping was conducted using MapChart. Calculation rates for synonymous (Ks) and non-synonymous (Ka) substitutions were estimated by Ka/Ks Calculator 2.0. Divergence time (T) was calculated using the formula $T = Ks/2\lambda$ ($\lambda = 1.5 \times 10^{-8}$) for millions of years ago (MYA) for dicotyledonous plants [38].

### 2.5. Intraspecific and Interspecific Collinearity Analyses

Collinearity analysis was completed using Circos software. The Multiple Collinearity Scan toolkit [38] was adopted to analyze gene duplication events with the default parameters. The syntenic analysis maps were built using multiple synteny plots to show the synteny relationship of the *P. trichocarpa* intra and interspecific DIRs.2.6. Plant cul-

tivation, treatments, and quantitative reverse transcription polymerase chain reaction (RT-qPCR) analysis.

In this study, the experimental plants used were *P. trichocarpa* and 'Nanlin 895'. *P. trichocarpa* was cultured on 1/2 standard wood plant medium [39]. 'Nanlin 895' (hybrid of *Populus deltoides* × *Populus euramericana*) was cultured on 1/2 Murashige and Skoog medium [40]. After one month of growth, seedlings were transferred to standard pots with a mixture of vermiculite/perlite/peat (1:1:3, v/v/v). *M. brunnea* was cultured in potato dextrose agar (PDA) at 25 °C [4,41]. After the second month of growth, at the 2-month-old seedling stage (15 cm high of aboveground part, 6 leaves, and about 4cm$^2$ of a single leaf), the two kinds of poplars were harvested to analyze gene expression in different tissues (root, stem, young leaf, and mature leaf) by RT-qPCR analysis. Two-month-old 'Nanlin 895' plants were pricked with one leaf (about 4cm$^2$) three punctures by a sterile needle and inoculated by spraying the leaves and stems thoroughly with a conidial suspension ($1 \times 10^6$ conidia per mL) harvested from 10-day-old PDA cultures. These plants were incubated in a greenhouse (60% humidity, 25 °C, 16/8 h day/night photoperiod, 20 μmol m$^{-2}$ s$^{-1}$). The plants were harvested immediately at eight time points: d 0, 0.5, and 1–6 after inoculation. At the same time, 'Nanlin 895' seedlings were treated with 200 μM abscisic acid (ABA), 10 mM salicylic acid (SA), 1 mM methyl jasmonate (MeJA), and 1 mM ethylene (ETH) before being harvested at 0, 3, 6, 9, 12, and 24 h for RNA extraction and RT-qPCR analysis [4,40]. Each experiment was replicated three times using three healthy plants. The specific primer sequences used for the RT-qPCR are listed in Supplementary Table S1. We use RNAprep Pure Plant Plus Kit (Tiangen, Beijing, China) to extract total RNA of respective tissues (starting 100 mg of each sample) according to the kit instructions. In this process, genomic DNA was removed by RNase-free DNase I. The concentration and purity of total RNA were determined by the spectrophotometer (drop, UK) and its integrity was tested by 1% agarose gel electrophoresis. 1 μg RNA of each sample was used to synthesize first-strand cDNA using HiScript$^®$ III 1st Strand cDNA Synthesis Kit (Vazyme, Nanjing, China). Then the cDNA was diluted 10-fold and was used for RT-qPCR. The RT-qPCR reactions were carried out by StepOnePlus$^{TM}$ (Applied Biosystems, USA) using a three-step PCR procedure with the following cycle parameters: 95 °C for 10 min, then 95 °C for 15 s and 60 °C for 1 min, for 40 cycles, and a melt cycle from 65 °C to 95 °C. The reaction mixture was 20 μL, including 2 μL cDNA, 0.4 μL primer-F (10 μmol/L), 0.4 μL primer-R (10 μmol/L), 10 μL ChamQ SYBR$^®$ qPCR Master Mix (Vazyme, Nanjing, China), 0.4 μL ROX, and 6.8 μL ddH$_2$O. The relative gene expression was calculated using $2^{-\Delta\Delta CT}$ method.

## 3. Results

### 3.1. Genome-Wide Identification and Characterization of DIR Proteins in Populus

A total of 50 DIRs were obtained by HMM analysis. After redundant sequences were removed, 40 sequences were reserved and submitted to the CDD, Pfam, and SMART to confirm the DIR domain. Finally, these 40 sequences were confirmed as *PtDIRs* and named for their positions on the chromosomes. Gene names, gene IDs, chromosomal locations, CDS lengths, intron numbers, amino acid numbers, molecular weights, pIs, subcellular locations, and N-glycosylation sites are listed in Table 1. The PtDIR proteins lengths ranged from 72 (PtDIR22) to 401 amino acids (PtDIR26). The molecular weights ranged between 7.57 kDa (PtDIR22) and 41.21 kDa (PtDIR26). The *PtDIRs* were distributed on 11 chromosomes. The predicted pI values ranged from 4.2 (PtDIR26) to 10.81 (PtDIR18). Almost all PtDIRs had at least one N-glycosylation site, except for PtDIR26 and PtDIR34. Subcellular localization prediction indicated that the PtDIRs were mainly located in the cell membrane, cell wall, and chloroplasts (chloro). A total of 28 PtDIRs (PtDIR1–3, 5–7, 10, 11, 13, 14, 16–28, 32–34, 36, and 38) were located in the cell membrane. Among these, five PtDIRs (PtDIR10, 13, 14, 16, and 32) were also located in the cell wall, ten (PtDIR1, 3, 5, 19, 20, 25, 28, 32, 36, and 38) were also located in the chloroplasts, and two (PtDIR3 and 24) were also located in the nucleus. Five PtDIRs (PtDIR4, 8, 9, 15, and 31) were specifically

located in the cell wall, and six (PtDIR12, 30, 35, 37, 39, and 40) were uniquely located in the chloroplasts. Finally, 25 PtDIRs (PtDIR1–11, 13, 14, 17, 19, 23–25, 27–30, 32, 35, and 36) had a signal peptide.

**Table 1.** Features of *PtDIRs* identified in *P. trichocarpa.*

| Name | Gene ID | Chr. | AA | MW (KDa) | pI | N-Glyc Number | Localization Predicted | Signal Peptide |
|------|---------|------|-----|----------|-----|---------------|------------------------|----------------|
| PtDIR1 | Potri.001G009100.1 | 1 | 185 | 20.38 | 10 | 2 | Cell membrane. Chloro | Yes |
| PtDIR2 | Potri.001G023600.1 | 1 | 189 | 20.88 | 9.79 | 3 | Cell membrane | Yes |
| PtDIR3 | Potri.001G023700.1 | 1 | 147 | 16.21 | 9.23 | 2 | Cell membrane. Chloro. Nucleus | Yes |
| PtDIR4 | Potri.001G023800.1 | 1 | 185 | 20.4 | 5.16 | 1 | Cell wall | Yes |
| PtDIR5 | Potri.001G054000.1 | 1 | 248 | 25.43 | 5.62 | 1 | Cell membrane. Chloro | Yes |
| PtDIR6 | Potri.001G054100.1 | 1 | 254 | 26.04 | 4.69 | 1 | Cell membrane | Yes |
| PtDIR7 | Potri.001G096500.1 | 1 | 182 | 20.61 | 6.94 | 4 | Cell membrane | Yes |
| PtDIR8 | Potri.001G096600.1 | 1 | 184 | 20.32 | 6.87 | 3 | Cell wall | Yes |
| PtDIR9 | Potri.001G096800.1 | 1 | 184 | 20.29 | 6.78 | 3 | Cell wall | Yes |
| PtDIR10 | Potri.001G096900.1 | 1 | 187 | 20.85 | 9.37 | 2 | Cell membrane. Cell wall | Yes |
| PtDIR11 | Potri.001G214600.1 | 1 | 178 | 19.03 | 9.07 | 6 | Cell membrane | Yes |
| PtDIR12 | Potri.002G131500.1 | 2 | 217 | 23.88 | 9.91 | 2 | Chloro | No |
| PtDIR13 | Potri.003G134400.1 | 3 | 187 | 20.76 | 9.1 | 3 | Cell membrane. Cell wall | Yes |
| PtDIR14 | Potri.003G134600.1 | 3 | 184 | 20.28 | 7.11 | 3 | Cell membrane. Cell wall | Yes |
| PtDIR15 | Potri.003G134700.1 | 3 | 215 | 24.47 | 9.8 | 3 | Cell wall | No |
| PtDIR16 | Potri.003G134800.1 | 3 | 199 | 22.22 | 8.45 | 3 | Cell membrane. Cell wall | No |
| PtDIR17 | Potri.003G174300.1 | 3 | 253 | 25.79 | 5.63 | 1 | Cell membrane | Yes |
| PtDIR18 | Potri.003G202000.1 | 3 | 94 | 10.76 | 10.8 | 2 | Cell membrane | No |
| PtDIR19 | Potri.003G216200.1 | 3 | 194 | 21.59 | 9.78 | 3 | Cell membrane. Chloro | Yes |
| PtDIR20 | Potri.003G216300.1 | 3 | 201 | 22.06 | 10 | 3 | Cell membrane. Chloro | No |
| PtDIR21 | Potri.003G216400.1 | 3 | 204 | 22.57 | 9.62 | 3 | Cell membrane | No |
| PtDIR22 | Potri.004G171400.1 | 4 | 72 | 7.57 | 4.67 | 2 | Cell membrane | No |
| PtDIR23 | Potri.005G100600.1 | 5 | 199 | 21.46 | 9.22 | 2 | Cell membrane | Yes |
| PtDIR24 | Potri.005G100700.1 | 5 | 197 | 21.01 | 9.77 | 2 | Cell membrane. Nucleus | Yes |
| PtDIR25 | Potri.006G195300.1 | 6 | 196 | 21.29 | 9.93 | 3 | Cell membrane. Chloro | Yes |
| PtDIR26 | Potri.008G049100.1 | 8 | 401 | 41.21 | 4.2 | 0 | Cell membrane | No |
| PtDIR27 | Potri.008G049200.1 | 8 | 311 | 32.46 | 4.48 | 2 | Cell membrane | Yes |
| PtDIR28 | Potri.008G061400.1 | 8 | 186 | 20.26 | 6.23 | 4 | Cell membrane. Chloro | Yes |
| PtDIR29 | Potri.009G130800.1 | 9 | 145 | 15.49 | 9.95 | 3 | Chloro. Golgi apparatus | Yes |
| PtDIR30 | Potri.009G130900.1 | 9 | 188 | 20.41 | 9.87 | 3 | Chloro | Yes |
| PtDIR31 | Potri.009G131000.1 | 9 | 207 | 22.36 | 9.56 | 4 | Cell wall | No |
| PtDIR32 | Potri.010G197000.1 | 10 | 188 | 20.74 | 9.78 | 4 | Cell membrane. Cell wall. Chloro | Yes |
| PtDIR33 | Potri.010G211800.1 | 10 | 313 | 32.49 | 4.58 | 2 | Cell membrane | No |
| PtDIR34 | Potri.010G211900.1 | 10 | 354 | 36.86 | 4.26 | 0 | Cell membrane | No |
| PtDIR35 | Potri.010G212000.1 | 10 | 240 | 25.62 | 9.71 | 3 | Chloro | Yes |
| PtDIR36 | Potri.016G060700.1 | 16 | 194 | 21.29 | 10.4 | 3 | Cell membrane. Chloro | Yes |
| PtDIR37 | Potri.016G060800.1 | 16 | 76 | 8.41 | 9.41 | 2 | Chloro | No |
| PtDIR38 | Potri.016G060900.1 | 16 | 197 | 21.55 | 10.3 | 3 | Cell membrane. Chloro | No |
| PtDIR39 | Potri.016G061000.1 | 16 | 195 | 21.32 | 10.1 | 3 | Chloro | No |
| PtDIR40 | Potri.018G125100.1 | 18 | 85 | 9.02 | 8.51 | 1 | Chloro | No |

### 3.2. Multiple Sequence Alignment, Phylogenetic Analysis, and Classification of PtDIRs

To clarify the evolutionary classification of the 40 *PtDIRs*, we constructed an evolutionary tree. The *DIR* gene family can be divided into six major subfamilies: DIR-a, b/d, c, e, f, and g (Figure 1). The *DIR* gene families of *P. trichocarpa* and *Arabidopsis* were only classified into three subfamilies (DIR-a, b/d, and e). Most *PtDIRs* belong to DIR-b/d (21 in total), followed by the DIR-a and DIR-e subfamilies with 8 and 11 *PtDIRs*, respectively. To date, studies have shown that the *DIR* genes, which can specifically control the oxidative coupling of coniferyl alcohol in the formation of (+) or (−) pinoresinol, were mainly concen-

trated in the DIR-a subfamily [11,12,35], and these functional genes are highlighted with a red background in Figure 1. However, GhDIR3 and GhDIR4 from *Gossypium hirsutum (G. hirsutum)*, which can contribute to forming the (P)-(+)-gossypol (dextrorotatory gossypol), belong to the DIR-b/d subfamily [36].

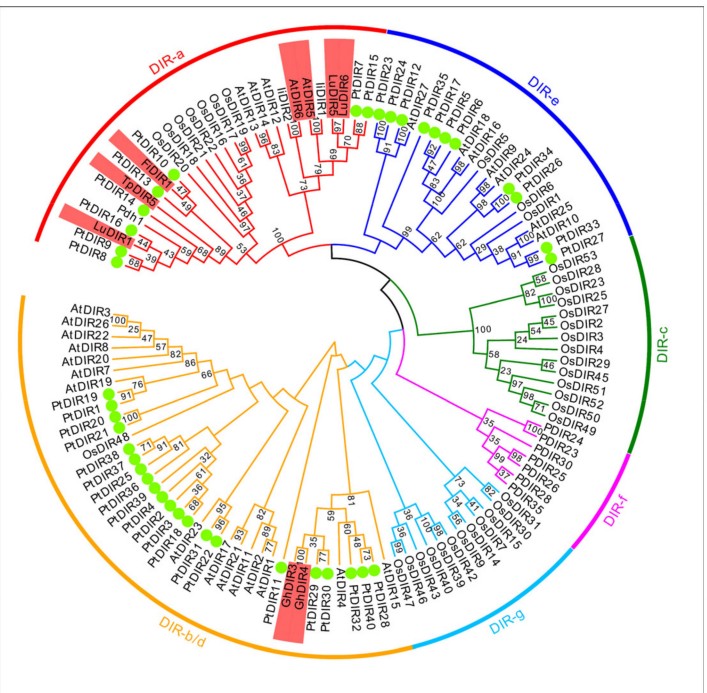

**Figure 1.** Phylogenetic relationship between *P. trichocarpa* and other plant species. *PtDIRs* are labeled with a green circle. The *DIR* genes that precisely control the stereoselective coupling of alcohol monomers in the formation of (+) or (−) pinoresinol are highlighted with a red background.

A comparison of PtDIR protein sequences shows that the protein similarity ranges from 3–95%, indicating that some PtDIR proteins may differ significantly and show functional diversity. In the DIR-a subfamily, the sequence similarity of PtDIR proteins is exceptionally high, ranging from 52–95%, followed by the DIR-e and DIR-b/d subfamilies, in which it ranged from 27–88% and 6–85%, respectively (Figure 2).

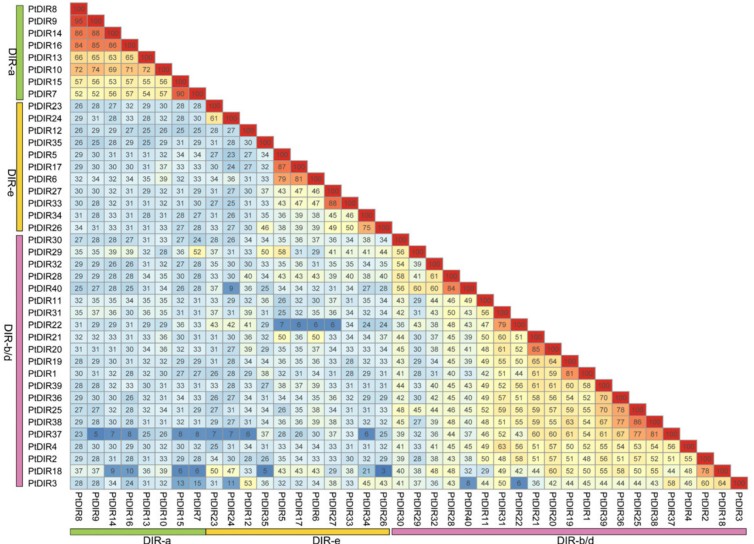

**Figure 2.** Identity matrix (percentage) of PtDIRs.

### 3.3. Gene Structure and Conserved Motifs of the PtDIR Gene Family

The conserved motifs of PtDIR proteins were identified by MEME, and ten putative motifs were revealed (Figure 3). The lengths of these predicted motifs ranged from 15–41 amino acids. The PtDIR proteins contained 2–7 conserved motifs. We selected the best possible matching sequences as the motif sequences (Supplementary Materials Table S2) and annotated each motif using CDD and Pfam software. Motif 2 was present in all PtDIR sequences except for PtDIR3. Motifs 1 and 3 were present in most DIR proteins. Motif 9 was only found in the proteins of group I and in the PtDIR2 protein of group V. Motifs 4 and 5 were only present in groups IV and V. Motifs 6 and 10 were present only in group I and motif 8 was found only in group V. The proteins from group II contain only two motifs (motifs 1 and 2). Group I is part of the DIR-a subfamily, groups II and III are part of the DIR-e subfamily, and groups IV and V belong to the DIR-b/d subfamily. The shared common motifs, seen via the distribution patterns of the various motifs across the different groups and within the same groups of PtDIR proteins in the phylogenetic tree, may indicate some functional diversity.

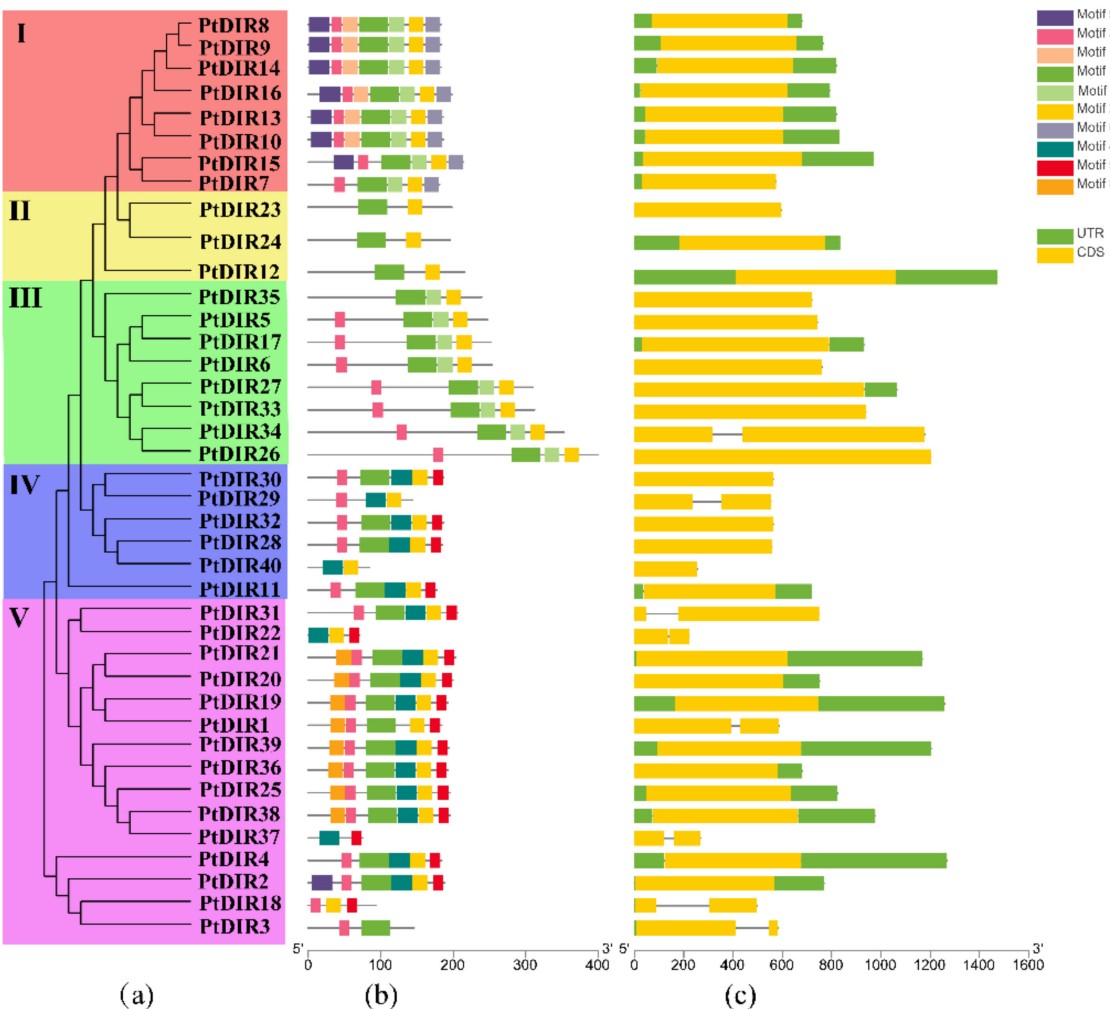

**Figure 3.** Phylogenetic relationship, conserved motif, and gene structure analyses of PtDIRs. (**a**) Phylogenetic tree of 40 PtDIRs. (**b**) Distribution of conserved motifs in PtDIRs. Ten putative motifs are shown in different colored boxes. (**c**) Exon/intron organization of PtDIRs.

The PtDIR protein structures were obtained from GSDS software. The results showed that all proteins belonging to the PtDIR family are stable. Most of the *PtDIR* genes (32/40, 80%) were intronless. In addition, the remaining eight PtDIR members (*PtDIR1, 3, 18, 22, 29, 31, 34,* and *37*) only contained one intron. Furthermore, approximately 42.5% of PtDIR

members did not have untranslated regions (UTRs). The PtDIR members from group I all contained 5′ and 3′ UTRs. In comparison, the PtDIR members from group IV had no UTRs, except for *PtDIR11*. The PtDIR members from the same branches often shared similar structures.

### 3.4. Chromosome Distribution and Ka/Ks

To examine the genome distribution of the *PtDIRs*, chromosomal mapping was performed by MapChart. As shown in Figure 4, 40 *PtDIRs* were distributed on 11 of the 19 *P. trichocarpa* chromosomes, indicating a diverse distribution. Chromosome 1 contains the highest number of *PtDIRs* (11), followed by chromosome 3 (nine *PtDIRs*). Chromosomes 2, 4, 6, and 18 only contain one *PtDIR* gene each. We calculated the Ka/Ks ratios of *PtDIR* gene pairs to understand the duplication events in the *PtDIR* gene family. A total of 14 *PtDIR* pairs were obtained, involving 18 *PtDIRs*. The divergence time estimates of the duplicated *PtDIR* gene pairs ranged from 3.44 MYA (*PtDIR8* and *PtDIR9*) to 35.54 MYA (*PtDIR10* and *PtDIR13*) (Supplementary Materials Table S3).

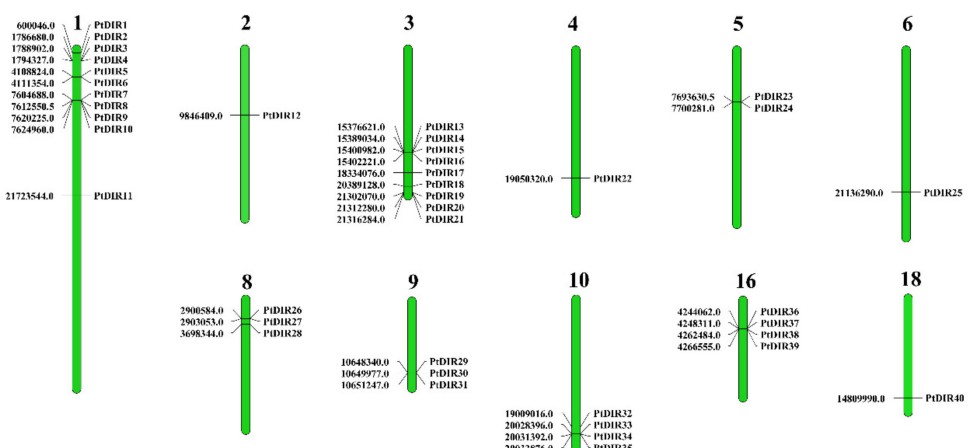

**Figure 4.** Chromosomal distribution of PtDIRs.

### 3.5. Stress-Related cis-Elements in the PtDIR Promoters

To further analyze the potential regulatory mechanisms of *PtDIRs* involved in plant stress responses, 12 *cis*-elements were selected and are shown in Figure 5. We found that 17 *PtDIRs* (*PtDIR2, 5, 12, 15, 17, 23–27, 29, 33–35, 37, 39*, and *40*) contained TC-rich repeats (defense and stress-responsive elements), indicating that the *PtDIR* gene family is related to responsiveness to biotic and abiotic stresses [42]. The phytohormone responsive elements ERE (ethylene), ABRE (ABA), CGTCA-motif (MeJA), TCA-element (SA), and the TATC-box (GA) are widely distributed in the *PtDIR* gene family. We found that 28 *PtDIRs* (1, 2, 5, 6, 8–12, 14, 16, 17, 20–24, 26–29, 32–34, 36–38, and 40), 25 *PtDIRs* (1, 3, 5–11, 14, 16–22, 24–26, 29, 34, 36, 38, and 40), 20 *PtDIRs* (1, 4, 6–8, 11–13, 16, 19, 20, 24, 27, 29, 31, 32, 35, 36, and 39), 17 *PtDIRs* (4, 6–8, 10, 11, 13, 14, 17–20, 31, 22, 34–36) and 5 *PtDIRs* (11, 15, 16, 23, and 33) contained ERE, ABRE, CGTCA-motif, TCA, and TATC-box *cis*-acting elements, respectively, suggesting that *PtDIRs* may participate in the regulation and metabolism of hormones in plants. Additionally, 19 drought-responsive elements (MBS), 14 low-temperature responsive elements (LTR), and 13 wound-responsive elements (Wunmotif) are also predicted to be specifically distributed in promoters of PtDIR*s*. Furthermore, four *PtDIRs* (29, 31, 34, and 36) have *cis*-acting MBSI elements, which are the MYB binding sites involved in flavonoid biosynthesis gene regulation. Some *cis*-acting elements like MRE, which is involved in light responsiveness, and the TGA-element, involved in auxin response, were also predicted, as shown in Supplementary Table S4. The *cis*-element analysis illustrated that *PtDIR* genes could respond to different stresses.

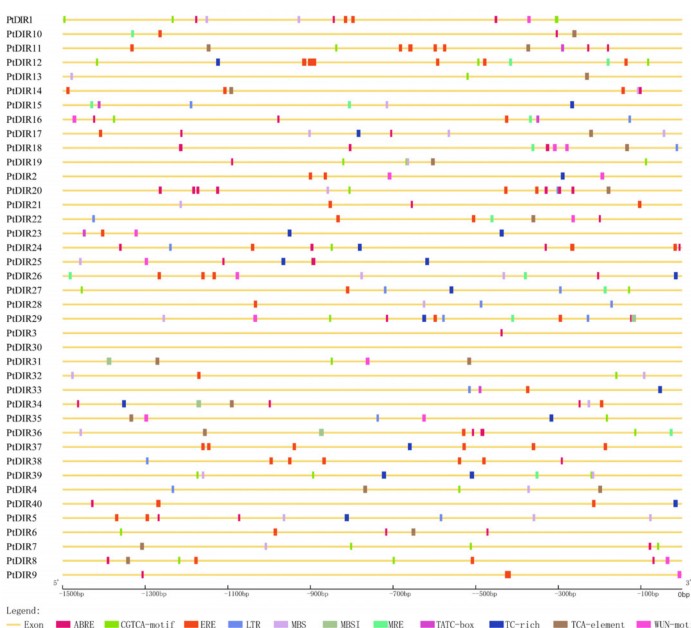

**Figure 5.** Predicted *cis*-elements in PtDIR promoters. Promoter sequences (−1500 bp) of 40 PtDIRs were analyzed by PlantCARE.

### 3.6. Synteny Analysis of PtDIRs in P. trichocarpa and Other Species

We conducted a synteny analysis on *PtDIRs* to investigate the duplication events occurring in the *PtDIR* gene family (Figure 6a). We identified 10 gene pairs (*PtDIR1/19*, *PtDIR1/36*, *PtDIR5/17*, *PtDIR8/13*, *PtDIR19/36*, *PtDIR19/25*, *PtDIR22/31*, *PtDIR25/36*, *PtDIR26/33*, and *PtDIR28/32*) that were clustered into 10 segmental duplication event regions on chr1, 3, 4, 6, 8–10 and chr16. Chr1 and chr3 contained three clusters, indicating that *PtDIRs* distributed on these two chromosomes had more connections. To further analyze the phylogenetic history of the *PtDIR* gene family in different species, we identified *DIR* gene family members using *Arabidopsis* as the dicot representative and rice as the monocot representative (Figure 6b). We identified 13 collinear gene pairs between *P. trichocarpa* and *Arabidopsis* and only two pairs between *P. trichocarpa* and rice. These collinear correlative genes are concentrated on chromosomes 1, 2, 3, 8, 9, and 10. *PtDIR26* and *PtDIR34* were anchored on both *Arabidopsis* and rice *DIR* genes. Four *PtDIRs* (7, 8, 15, and 34) were respective collinear with two different *AtDIR* genes simultaneously, indicating potential paralogous gene pairs, and they may have played an essential role in the *DIR* gene family during evolution.

### 3.7. Analysis of Protein Tertiary Structures and Sequence Alignments of PtDIR Proteins of the DIR-a Subfamily

Phylogenetic analysis revealed that eight PtDIRs (PtDIR7–10 and PtDIR13–16) were DIR-a subfamily members. In this study, using AtDIR6 associated with (−) pinoresinol and PsDRR206 associated with (+) pinoresinol as references, we predicted the 3D structures of eight PtDIR proteins of the DIR-a subfamily. The 3D structures of PtDIRs 7 and 15 and PtDIRs 8–10, 13, 14, and 16 were compared and merged with AtDIR6 and PsDRR206, respectively, using Chimera software. The 3D structures of the PtDIR proteins were highly similar to AtDIR6 or PsDRR206. Integration analysis of the conserved domains (red) showed that those of the PtDIRs could be integrated with AtDIR6 and PsDRR206 (Figure 7a).

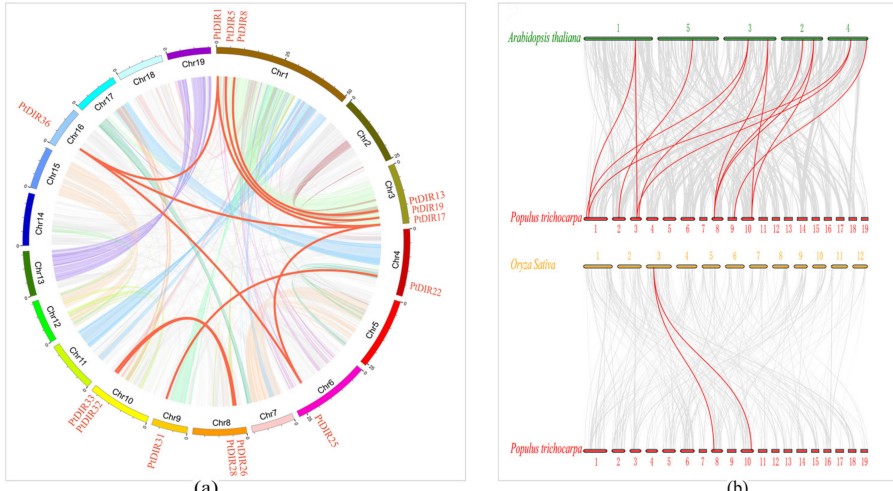

**Figure 6.** Synteny analysis of PtDIRs. (**a**) Schematic representations of the interchromosomal relationships of PtDIRs. The red lines indicate duplicated PtDIR gene pairs. (**b**) Synteny analysis of DIR genes among Populus, Arabidopsis, and rice. Gray lines in the background show the collinear blocks within *P. trichocarpa* and other plant genomes; the red lines highlight the syntenic DIR gene pairs.

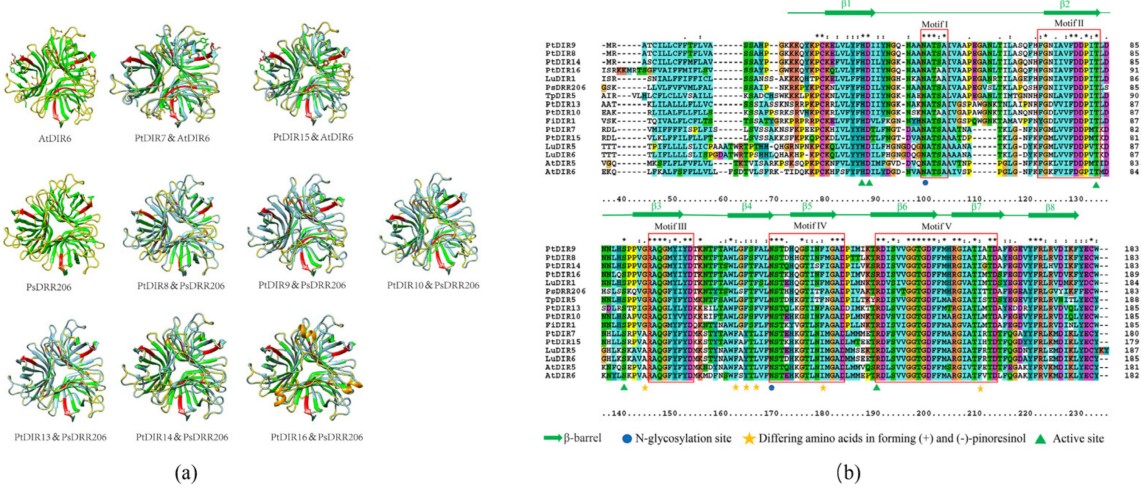

**Figure 7.** The protein tertiary structures and sequence alignments of PtDIR proteins of the DIR-a subfamily. (**a**) The prediction of the PtDIR7–10, 13, 15, and 16 protein tertiary structures of the DIR-a subfamily compared and merged with AtDIR6 (associated with (−) pinoresinol) and PsDRR206 (associated with (+) pinoresinol), respectively. (**b**) The alignment of PtDIR protein sequences of the DIR-a subfamily.

Next, we performed a protein sequence alignment using ClustalX 2.0 (Figure 7b). The DIR proteins included FiDIR1, PsDIR202, TpDIR5, TpDIR8, and LuDIR1, which can contribute to forming (+) pinoresinol, and AtDIR5, AtDIR6, LuDIR5, and LuDIR6, which can contribute to forming (−) pinoresinol. The result showed that they have high similarity, with five conserved motifs. Motif I was located from amino acid positions (aa) 100–105. Motifs II–V were located from aa 123–134, 146–154, 170–184, and 191–215, respectively. All protein sequences contained an eight-stranded antiparallel β-barrel (green arrow), five active sites (His87, Asp88, Thr134, Ser141, and Arg191; green triangles), and two N-glycosylation sites (Asn; blue circles). Six differentially conserved residues at aa 145, 162, 165, 167, 180, and 211 were involved in forming (+) pinoresinol or (−) pinoresinol (yellow star). Residues 145, 167, and 180 of *PtDIRs* are accordant with other *DIR* genes; however, the residue 162 of PtDIR13 is Phe instead of Leu, and residue 211 of PtDIR13 and PtDIR10 is Leu, not Ile. The alignment results suggest that PtDIR8–10, 13, 14, and

16 may catalyze the formation of (+) pinoresinol, with PtDIR8, 9, 14, and 16 showing the most potential. Residue 211 of PtDIR15 and PtDIR7 is Phe instead of Tyr, while the other residues are conserved. It is likely that PtDIR7 and PtDIR15 can catalyze the formation of (−) pinoresinol rather than (+) pinoresinol.

### 3.8. Different Tissue Expression Patterns of DIR Genes in P. trichocarpa and 'Nanlin 895'

The 'Nanlin 895' poplar is widely planted in the middle and lower reaches of the Yangtze River in China. Compared with other local poplar species, it shows superior growth, and has high transgenic success rate. So *'Nanlin 895'* is a good material to study the gene function of poplar [43]. This makes the 'Nanlin 895' poplar ideal for the study of gene functions in poplars. We used RT-qPCR to analyze *DIR* gene expression levels in root, stem, young leaf, and mature leaf tissues in *P. trichocarpa* and 'Nanlin 895'. The results are presented in the form of heatmaps, and the expression trends are clustered. As shown in Figure 8, the tissue expression patterns of *P. trichocarpa* and 'Nanlin 895' show similarities and differences. Most *PtDIRs* and 'Nanlin 895' *DIR* genes (*PeDIRs*) were expressed in all tissues, except for *PtDIR29*, *30*, and *40* and *PeDIR29*, *30*, and *40*, which were barely expressed in any tissues. *Pt/PeDIR3* and *Pt/PeDIR37* showed no expression in root and leaf tissues, respectively. *Pt/PeDIR28* was expressed specifically in the stem. *PeDIR2* was not expressed in roots, while *PeDIR22* was specifically expressed in roots and *PtDIR2* and *PtDIR22* were expressed in all tissues. We found that 19 *PtDIRs* (5–9, 11, 13, 15–18, 20, 23, 26, 27, 31, 33, 35, and 36) and 16 *PeDIRs* (5, 6, 8, 9, 12, 15–18, 22, 28, and 31–35) were relatively highly expressed in roots, 14 *PtDIRs* (1–3, 10, 12, 14, 19, 21, 24, 28, 34, and 37–39) and 14 *PeDIRs* (1, 7, 10, 11, 13, 14, 19, 21, 24, 26, 27, 37, 39) were relatively highly expressed in stems, and four *PtDIRs* (4, 22, 25, and 32) and seven *PeDIRs* (2–4, 23,25, 36, and 38) were relatively highly expressed in leaves.

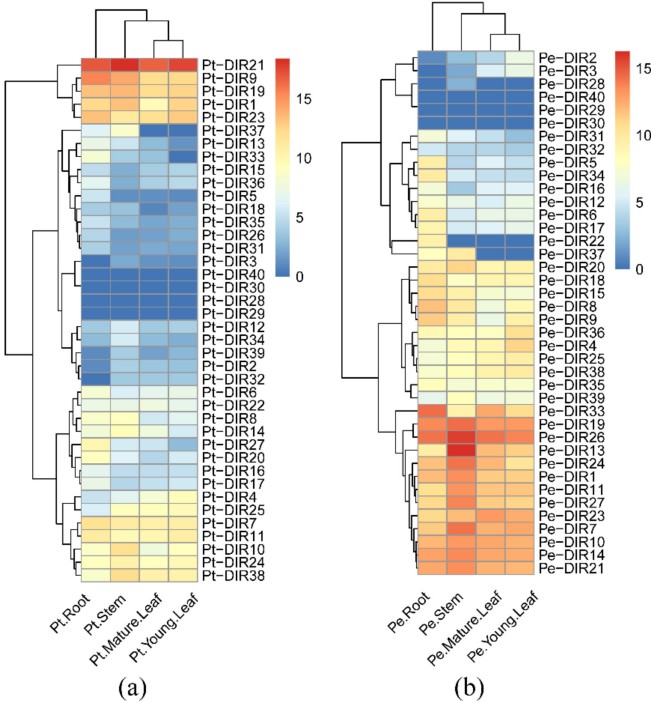

**Figure 8.** Expression profiles of DIR genes in various tissues (root, stem, young leaf, and mature leaf) of *P. trichocarpa* and 'Nanlin 895' by heatmap. (**a**) Tissue-specific expression of PtDIRs in *P. trichocarpa*. (**b**) Tissue-specific expression of PeDIRs in 'Nanlin 895'. Data represent mean values ± the standard error (SE) of at least three independent biological replicates for RT-qPCR.

### 3.9. Expression Profiles of 'Nanlin 895' PeDIRs in Response to M. brunnea

To profile the responses of *PeDIR*s to *M. brunnea*, we sprayed 2-month-old 'Nanlin 895' poplars with the bacterial suspension solution. Samples were collected at eight time points (d 0, 0.5, and 1–6) and the expression patterns of 37 *PeDIR*s were analyzed by RT-qPCR using AB073011 (actin) as the housekeeping gene. As shown in Figure 9, the *PeDIR*s that responded to *M. brunnea* were mainly in the DIR-a and DIR-b/d subfamilies. Except for *PeDIR*17 and *PeDIR*23, the genes in the DIR-e subfamily showed no apparent response to *M. brunnea*. The expression profiles of seven *PeDIR*s (7–10, 13-15) in the DIR-a subfamily, 10 *PeDIR*s (1, 11, 19–21, 25, 36, and 37–39) in the DIR-b/d subfamily, and two *PeDIR*s (17 and 23) in the DIR-e subfamily showed more than two-fold increased expression for at least one time point compared to the d 0 treatment. Among the 19 *PeDIR*s that showed strong induction by *M. brunnea*, the expression levels of eight *PeDIR*s (7–10, 15, 19, 21, and 38) peaked on the third day after inoculation. The expression levels of two *PeDIR*s (13 and 25) peaked at d 0.5 after inoculation. *PeDIR16*, *33* and *35* showed decreased expression compared to d 0 treatment. Notably, *PeDIR11* and *PeDIR36* of the DIR-b/d subfamily showed a significant increase of more than seven-fold compared to d 0.

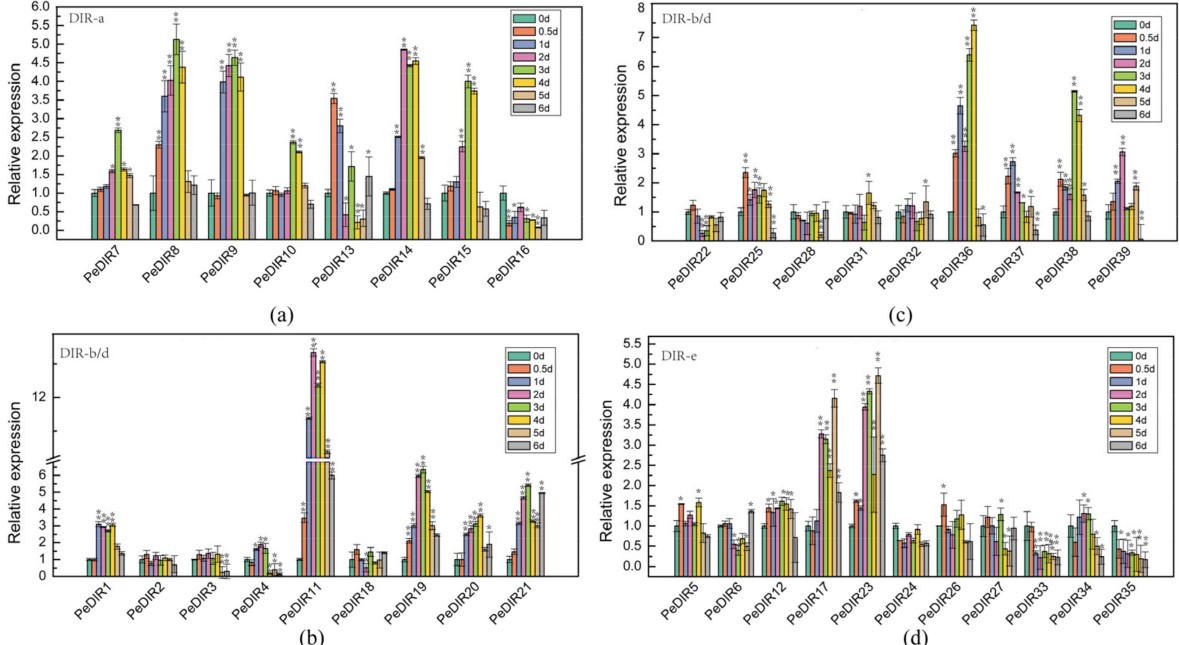

**Figure 9.** Relative expression levels of the PeDIRs of 'Nanlin 895' seedlings subjected to *M. brunnea* for 0, 0.5, and 1–6 d. Expression of the PeDIRs normalized to those of actin and shown relative to the expression at 0 h. The $2^{-\Delta\Delta Ct}$ method was used to compute the expression levels of PeDIRs at different times. Experiments were repeated at least three times. (**a**) Relative expression levels of PeDIRs of the DIR-a subfamily in response to *M. brunnea*. (**b**,**c**). Relative expression levels of PeDIRs of the DIR-b/d subfamily in response to *M. brunnea*. (**d**) Relative expression levels of PeDIRs of the DIR-e subfamily in response to Marssonina brunnea. Statistically analyzed using student's t-tests (* *p* < 0.05, ** *p* < 0.01).

### 3.10. Expression Patterns of PeDIRs of 'Nanlin 895' in Response to Different Phytohormones

To further analyze the response patterns of *PeDIR*s to phytohormones, we selected 19 *PeDIR*s from three *PeDIR* subfamilies that did not only respond to *M. brunnea* stronger than others but also showed a higher expression in tissues, and further we explored their response patterns using four hormones (ABA, SA, MeJA, and ETH; heatmap of Figure 10 and bar chart of Figure S1). We included seven *PeDIR* genes from DIR-a (7–10, 13–15), ten from DIR-b/d (1, 11, 19-21, 25, 36, and 37–39), and two from the DIR-e subfamily (17 and 23) (Figure 9). Some *PeDIR*s, such as *PeDIR9* and *PeDIR21*, were upregulated by all four phytohormones, and, except for *PeDIR21* and *PeDIR37*, the remaining *PeDIR*s showed a

strong response (more than 2-fold higher expression at least one time point than the d 0 after treatment) to ABA. *PeDIR8* and *39* only showed strong responses to ABA, SA, and MeJA, while *PeDIR11*, *14*, and *25* only showed strong responses to ABA, SA, and ETH. *PeDIR20* showed an obvious response to ABA, MeJA, and ETH, and *PeDIR15* showed a strong response to ABA and SA, while *PeDIR7* and *PeDIR19* showed strong responses to ABA and MeJA. *PeDIR10*, *36*, *37*, and *38* were downregulated under SA stress but upregulated under ABA, ETH, and MeJA stresses.

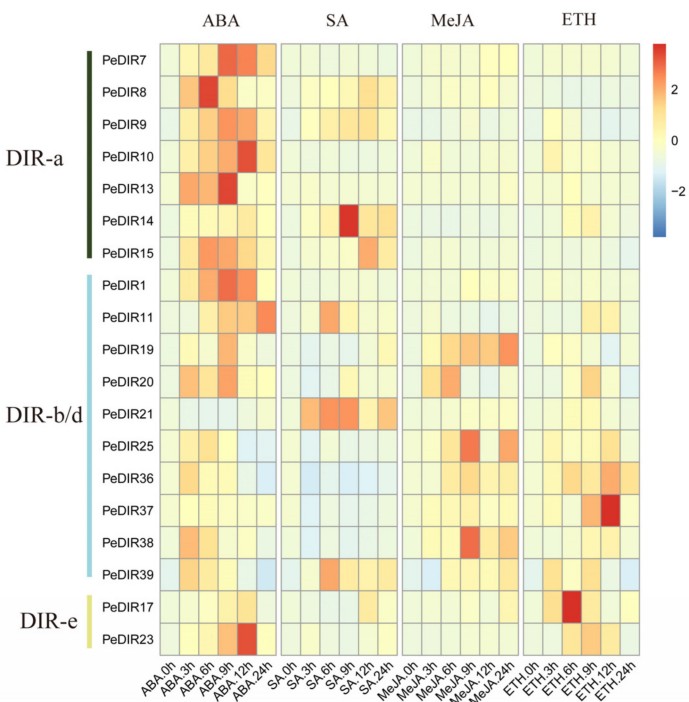

**Figure 10.** Hormone response pattern analysis of PeDIR genes by heatmap. PeDIR expression in the 'Nanlin 895' response to exogenous hormone (ABA, SA; MeJA, and ETH) treatment for 0, 3, 6, 9, 12, and 24 h.

## 4. Discussion

DIR proteins are widely distributed in terrestrial plants [23,28,35,44,45]. However, there are no reports of DIR proteins in primitive aquatic plants. Therefore, it is speculated that DIR proteins are associated with the development of vascular bundles that occurred during the evolution of aquatic plants to terrestrial plants [46], and they likely play an important role in the formation of lignin and lignan. They are involved in a broad range of plant biological functions such as resistance to various biotic and abiotic stresses. Previous studies identified 19, 49, 24, 29, and 35 *DIR* genes in *I. indigotica*, rice, pepper, pear, and brassica, respectively [15,28,35,47,48]. To date, little is known about the *DIR* family of genes in *P. trichocarpa*. In this study, we identified 40 *DIR* genes in the *P. trichocarpa* gene database. These genes were distributed on 11 chromosomes, and 80% *PtDIRs* were intronless. N-glycosylation is important for secretory proteins [49] and also is a vital characteristic of DIR proteins [11]. Approximately 95% of PtDIR proteins have at least one N-glycosylation site. The signal peptide can direct the newly-synthesized protein to various subcellular organelles [50]. Around 62.5% of PtDIR proteins contain signal peptide sequences, and our subcellular localization results showed that PtDIR proteins were located in various organelles, indicating that PtDIR proteins can be targeted for extracellular release before finally localizing in various subcellular organelles.

Phylogenetic analysis of *DIR* genes from multiple species showed that they could be divided into six subfamilies (DIR-a, b/d, c, e, f, and g), but not every species has all of the subfamilies [23]. For example, brassica only has the DIR-a and DIR-g subfamilies [28],

DIR-f is exclusively found in spruce [23], and no angiosperm DIR proteins were observed in the DIR-f subfamily [23]. In addition, the DIR-c subfamily is only found in angiosperm monocots. The *DIR* genes of the DIR-c subfamily always combine with the jacalin-like domain to contribute to the defense against insects [15,51,52]. *P. trichocarpa* has three subfamilies (DIR-a, b/d, and e). The DIR-a subfamily has eight *PtDIRs*, and compared to other *DIR* genes that can control the oxidative coupling of coniferyl alcohol specifically in the formation of (+) or (−) pinoresinol, these eight *PtDIRs* can be divided into two groups. One group (PtDIR8–10, 13, 14, and 16) possibly contributes to the formation of (+) pinoresinol, while the other (DIR7, 15) possibly contributes to the formation of (−) pinoresinol. This was confirmed following analyses of the tertiary structures, conserved motifs, and protein sequence alignments. This is similar to flax in which LuDIR1, 5, and 6 belong to the DIR -a subfamily. It is known that LuDIR5 and LuDIR6 contribute to forming (−) pinoresinol in seed-coats, while LuDIR1 contributes to forming (+) pinoresinol in the vegetative organs [44,53]. However, not all the *DIR* genes of DIR-a can form a single chiral pinoresinol [13,44]. Further in vitro and in vivo experiments are needed to verify the ability of PtDIRs (especially the DIR-a subfamily) to form unique chiral pinoresinol.

The *cis*-elements in the 5′ upstream region are involved in the dynamic regulation of gene expression [54]. *PtDIRs* contain elements related defense and stress responses, indicating that they may play an important role in biological and abiotic stresses, and a large number of elements are responsive to hormones such as ABA, SA, MeJA, and ETH. To be able to respond to various stresses, plants have a complex signal transduction network and synthesize a class of small hormones (SA, MeJA, ETH, and ABA) as signaling molecules in the plant defense response [55–57]. This indicates that the expression patterns of *PtDIRs* closely rely on the regulation of the hormone network. The presence of duplications in the genome helps plants adapt to the changing environment and evolve more smoothly [58]. All tandem duplicated *PtDIR* gene pairs had a Ka/Ks < 0.5, indicating that the *PtDIR* gene family might have undergone strong purifying selective pressures during evolution [59]. It is estimated that the divergence time of duplicated *PtDIR* gene pairs from 3.44 (PtDIR8 and PtDIR9)–35.5 (PtDIR10&PtDIR13) MYA, indicating *PtDIRs* had undergone duplication events over a long time span. Duplicated genes provide a template for new genes that take on new functions [60]. *P. trichocarpa* contains 10 segmental duplication event regions distributed on eight chromosomes, indicating that segmental duplication events were crucial in the expansion of the *PtDIR* gene family. Synteny analysis of *P. trichocarpa* with *Arabidopsis* and rice showed that *P. trichocarpa* might be more closely related to *Arabidopsis*. *PtDIR26* and *PtDIR34* were anchored in both rice and *Arabidopsis*, indicating that these collinear pairs may have existed before the evolution of monocotyledons and dicotyledons.

The different tissue expression patterns of the *DIR* genes have been described in many species, such as pepper, flax, rice, and brassica [15,28,44,47,61]. There is no unified tissue expression pattern for *DIR* genes. The 'Nanlin 895' poplar has a mature genetic transformation system and is an ideal model to study gene function in poplar [43,62], thus we used 'Nanlin 895' alongside *P. trichocarpa* to analyze *DIR* genes. The expression profile analysis of *DIR* genes in *P. trichocarpa* and 'Nanlin 895' in different tissues (root, stem, young leaf, and old leaf) showed similarities and differences. In *P. trichocarpa*, there were 19, 14, and 4 *PtDIRs* that were highly expressed in root, stem, and leaf tissues, respectively, while there were 16, 14, and 7 *PeDIRs* that were highly expressed in 'Nanlin 895' root, stem, and leaf tissues, respectively. Among them, 11, 8, and 2 *Pt/PeDIRs* had similar expression patterns in roots, stems, and leaves, respectively. The result revealed gene expression diversity among tissues and species. In pepper, the expression of most *CaDIR* genes was lowest in the roots [47]. The tissue expression in poplar differs from the *CaDIR* genes of pepper but is consistent with the *DIR* genes from *Brassica rapa* and *I. indigotica* [28,35]. GhDIR3 and GhDIR4 are important for the coupling of galactosyl and contribute to producing (+) gossypol [36]. Based on the analysis of the evolutionary tree, *GhDIR3* and *GhDIR4* are closely related to *PtDIR28–30, 32*, and *40*, but *Pt/PeDIR29, 30*, and *40* were not detected in either poplar, and the expression of *Pt/PeDIR28* was very low in several tissues. Only

*Pt/PeDIR32* was highly expressed in various tissues. Based on these results, we inferred that *Pt/PeDIR32* has a special function in poplar. In *Arabidopsis*, AtDIR10, which is highly expressed in roots, is essential for the correct formation of the Casparian strips in the roots [63]. The related genes in poplar are *Pt/PeDIR27* and *Pt/PeDIR33*. *Pt/PeDIR33* and *PtDIR27* were also specifically expressed in roots, indicating that *Pt/PeDIR33* and *PtDIR27* also play an important role in root development. Lignification of vegetative organs is essential for healthy plant growth, especially for vascular plants [20]. Previous studies have shown that *DIR* genes can contribute to the lignification of plant tissues. For example, the seed coat of *Arabidopsis* [64], the pod wall of soybean [65], the hypocotyl of hemp [66], the seed of flax [53], and the cell stone of pear [48] contain traces of the lignification function of the *DIR* genes. The *DIR* genes of *P. trichocarpa* and 'Nanlin 895' were diversely expressed, indicating the probable function of *Pt/PeDIRs* in lignification during tissue development or biological and abiotic stresses.

Studies and analyses of plant disease resistance genes have always been essential [67,68]. Poplar is a perennial, tall, woody plant and is more susceptible to pathogen stress than common herbs in both time and space [5,69]. Previous studies have discussed the role of DIR genes in disease resistance in cotton [18], pepper [47], soybean [17], and wheat [16]. In 'Nanlin 895', 19 *PeDIR*s (7, 10, and 2 *PeDIR*s belonging to DIR-a, DIR-b/d, and DIR-e, respectively) were able to respond to *M. brunnea* stress, showing expression levels more than two-fold higher than the treatment at d 0. The *M. brunnea*–responsive *PeDIR*s were mainly distributed across the DIR-a and DIR-b/d subfamilies. It is likely that the DIR-e subfamily has other functions. Most *PeDIR*s showed no significant changes at d 0.5. This result is similar to that of Yuan's study concerning the response of other genes to *M. brunnea*, which indicated that the gene expression profiles showed slightly increased expression at the early stage, within 12 h after inoculation [5]. Most *PeDIR* responses to the pathogen occurred rapidly at d 1–4 following inoculation and then remained stable at d 5 and 6. In *V. vinifera* infected with *E. necator*, analysis of the coniferyl alcohol content changes at different stages led Borges et al. to hypothesize that low *DIR* expression in the early stages might initially contribute to lignin biosynthesis, and then high *DIR* gene expression at later stages could increase lignan biosynthesis to enhance antimicrobial activities of plants [33]. It is possible that 'Nanlin 895' also has a similar disease-resistance pattern. Plant defense responses to pathogens are complex. Several important signaling molecules are involved in disease resistance regulation, including SA, JA, ABA, and ETH [6,10,70]. Usually, the plant genes can respond rapidly following a variety of phytohormone treatments. The plant hormone signal transduction network is a key regulatory system for the interaction between plants and pathogens [71,72]. In 'Nanlin 895' we analyzed the responses of 19 *PeDIR*s to phytohormones, and those responses were complex. Previous studies have shown that under *M. brunnea* induction, *PdLOX1* and *PdLOX2* genes in poplar can respond strongly [4]. The expression of *PdLOX1* and *PdLOX2* is downregulated under SA stress but upregulated under MeJA stress [4]. It is speculated that *PdLOX1* and *PdLOX2* genes may confer disease resistance mainly through the MeJA signaling pathway. In addition, other research also showed that the *AOX* gene could respond to *M. brunnea* stress. The expression of the *AOX* gene was downregulated under SA stress but upregulated under MeJA and ETH stresses [6]. However, other studies showed that *WRKY* genes could respond to stress and were upregulated in response to SA stress [9]. Under *M. brunnea* induction, poplar *PPR* gene expression was also upregulated under SA and MeJA [73]. We showed that the plant hormone signaling pathways are not independent, and complex crosstalk takes place. These results suggest that *PeDIR*s may play important roles in disease resistance in poplar. Further research is needed on the specific mechanisms of disease resistance and their association with various hormone pathways.

## 5. Conclusions

We identified 40 *PtDIRs* from the *P. trichocarpa* genome. They were characterized and further classified into three subfamilies of DIR-a, DIR-b/d, and DIR-e, and the motifs and

protein sequences were conserved within each subgroup. Gene duplication and calculation of Ka/Ks values, alongside synteny analysis of intra- and interspecific collinearity, provided an insight in the evolutionary history of the *PtDIR* genes. The tertiary structures of the PtDIR proteins from the DIR-a subfamily showed that PtDIR7 and 15 could possibly catalyze the formation of (−)-pinoresinol, while PtDIR8, 9, 10, 14, and 16 could possibly catalyze the formation of (+)-pinoresinol. The *cis*-acting elements of *PtDIRs* were involved in the response to biotic stress, abiotic stress, and various phytohormones, including ABA, SA, MeJA, and ETH, implying that *PtDIRs* might be involved in stress resistance. The expression patterns seen in root, stem, young leaf, and mature leaf tissues in *P. trichocarpa* and 'Nanlin 895' revealed the diversity of *DIR* gene expression. The expression patterns of the *PeDIR*s in response to *M. brunnea* and various phytohormones (such as ABA, SA, MeJA, and ETH) confirmed the disease resistance roles of the *DIR* genes and the complex phytohormone signaling pathway. These results provide valuable evidence for better understanding of the biological role of *DIR* genes in poplar.

**Supplementary Materials:** The following are available online at https://www.mdpi.com/article/10.3390/f12040507/s1, Data S1: Text file of the alignment corresponding to the phylogenetic tree in Figure 1, Table S1: List of primers for the RT-qPCR analyses, Table S2: Sequences of 10 predicted motifs of the PtDIR proteins, Table S3: Ka/Ks analysis of *PtDIR* gene pair duplications, Table S4: Transcription factors predicted for the binding promotors of the *PtDIR* genes.

**Author Contributions:** L.L. has written the draft and prepared the manuscript; W.S., P.Z. have designed the experiments; H.W., P.W., H.L., S.R. and D.L. have reviewed and modified the manuscript; Q.Z. have supervised the manuscript; Q.Z. has funded this project. All authors have read and agreed to the published version of the manuscript.

**Funding:** This research was supported by the National Natural Science Foundation of China (NSFC) (31570650), and the Priority Academic Program Development of Jiangsu Higher Education Institutions.

**Data Availability Statement:** All datasets presented in this study are included in the article/ Supplementary Materials.

**Conflicts of Interest:** The authors declare no conflict of interest.

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
