# Peer review of "Genome-Wide Characterization of Dirigent Proteins in Populus: Gene Expression Variation and Expression Pattern in Response to Marssonina brunnea and Phytohormones"

_forests, doi:10.3390/f12040507_

Round 1

Reviewer 1 Report

This manuscript was about populous DIR gene family analysis. My concerns are as follows:

  1. The language of this manuscript need to be polished by native English speaker or language editing companies.
  2. For Fig 8 and 9, please provide statistical analysis for the gene expression.
  3. It’s hard to see the x and y axis of the table in Figure 9. I suggest the authors to present them clearly or show the expression profile in heat map.

Author Response

Dear professor:

Thanks for your valuable suggestions and comments to our last submitted manuscript.  We have modified the manuscript according to the your suggestions.

Point 1: The language of this manuscript need to be polished by native English speaker or language editing companies.

Response 1: Thanks for your valuable comment. The English in this document has been checked by at least two professional editors, both native speakers of English. For a certificate, please see: http://www.textcheck.com/certificate/31ueHQ. What’s more, we also accepted detailed suggestions about language modification from all of the reviewer, and the modification has been completed accordingly. 

Point 2: For Fig 8 and 9, please provide statistical analysis for the gene expression.

Response 2: Thanks indeed for your valuable comment. The statistical analysis in Fig 8 (Fig 9 according to corrected numeration) and Fig 9 (Fig S1 according to corrected numeration) for the gene expression have been provided accordingly.

Point 3: It’s hard to see the x and y axis of the table in Figure 9. I suggest the authors to present them clearly or show the expression profile in heat map.

Response 3: Thanks for your valuable suggestion. Fig 9 (Fig 10 according to corrected numeration) has been shown by heatmap, and the bar gragh data of Fig 9 has been removed to the supplementary file: Fig S1.

Reviewer 2 Report

Dear authors,

Thank you for the interesting work and high quality of the manuscript. I enjoyed reading it.

Overall, I consider your work suitable for publication after minor changes, most serious of them is about the use of only a single endogenous control for RT-qPCR (I provided a strategy to resolve this).

Please consider the following suggestions and comments (line numbers indicated as in the pdf file provided to me):

Line 12 – replace “Marssonina brunnea is” with “Marssonina brunnea causes”;

Line 17 – delete “further”

Line 18 – substitute “We found 40 PtDIR” with “These”

Line 18 – delete “(PtDIRs)”

Line 19 – change “38/40” to “32/40”

Line 26 – use P. trichocarpa, not the full name

Lines 27, 29, 31 – use M. brunnea, not the full name

Line 30 – insert “also” after “PeDIRs”

Line 30 – substitute “after treatment with” with “in response to”

Line 30 – change “suggested” to “suggest”

Line 40 – delete “(M. brunnea)”

Line 41 – delete “leaf”

Line 43 – insert reference after “1997” (even if it’s the reference 7, please insert the reference here as well)

Line 51 – substitute “present” with “localization”

Line 51 – substitute “was” with “were”

Line 66 – insert “responding to them” after “[25]”

Line 79 – substitute “members” with “genes”

Line 143 – use “RT-qPCR” instead of “qRT-PCR”. This is the more common abbreviation for Quantitative Reverse Transcription Polymerase Chain Reaction. Also, instead of “real-time quantitative polymerase chain reaction” use “quantitative reverse transcription polymerase chain reaction”.

Line 150 – you probably mean “seedling”

Line 152 – 153 – Please explain pricking in more detail – how many punctures per a certain area were performed?

Line 154 – 155 – Please mention the details about the lighting mode

Line 156 – Please clarify if the time point “d 0” represents a non-inoculated control or a sample collected immediately after inoculation

Line 158 - Please clarify if the time point “0 h” represents a non-treated control or a sample collected immediately after treatment

Ending of 2.6. section – please indicate the kits and mention the main details regarding RNA extraction (starting amount of sample material for RNA extraction for each tissue type) and RT-qPCR (reaction mixture, amount of RNA used per reaction and thermal cycling protocols; did you use a one-step or a two-step protocol) even if you have referred to publications describing these procedures. If prior RT-qPCR you have determined the RIN values, please mention them.

Line 192 – insert reference after “DIR-a subfamily”

Table 2 – In my opinion, because of the looks of it, the Table 2 should be considered to be a Figure (Figure 2). Please change Table 2 to Figure 2 and the numeration of following figures accordingly.

Line 211 – delete “a range of”

Line 251 – please insert a reference following “(defense and stress-responsive elements)” describing TC-rich repeats as defense and stress-responsive elements.

Line 258 – 259 – please elaborate. You listed the phytohormone-responsive elements. This proves that these PtDIRs respond to these phytohormones. What makes you think that these genes are also involved in regulation and metabolism of these phytohormones? What additional evidence / references can you offer?

Line 262 – please insert “promoters of” between “in” and “PtDIRs”

3.5. section in general – why did you focus specifically on a 1500 bp long sequence for cis-element search? Please provide a reference or other information supporting this decision.

Line 279 – use “10 gene pairs” instead of  “20 genes”

Line 284 – use “history” instead of “mechanisms”

General remark about Figure 6 (Figure 7 according to corrected numeration) – I hope that this figure will be available as a high-resolution image. Please also provide an explanation for the green arrows in B part of the Figure, just like you do for the other symbols.

Line 330 – what do you mean by “has a mature genetic transformation system”? Could you rephrase?

Line 353 – delete “23” from the word “collected”

Remark about actin as an endogenous control – Usually more than 1 endogenous control is necessary for the results of gene expression to be published. Despite actin being a housekeeping gene, the MIQE Guidelines (Minimum Information for Publication of Quantitative Real-Time PCR Experiments) strongly advise against using a single endogenous control. Do you have supporting information showing that in this case actin works fine? Assuming that you started with the same amount of RNA in each reaction (cDNA if you used a 2-step protocol) I suggest that you include the data about variation amplitude of the Ct values for the actin gene expression. If you can do this, it should be enough. If you haven’t used the same amount of RNA / cDNA for the gene expression reactions, provide a the information about the correlation of actin Ct values to the amount of RNA / cDNA template.

Line 361 – use “Among” instead of “In”

Line 362 – substitute “disease resistance” with “induction by M. brunnea”

Line 363 – delete “treatment”, delete “only”

Line 370, 373, 374, 375 – use M. brunnea, not the whole name

Line 383 – substitute “could” with “did”

Line 387 – use a comma before “except”

Line 388 – for this time only, please explain in brackets following the words “strong response” what exactly are the criteria you use to define a strong response

General note 1 – In discussion or introduction please mention which phytohormones are crucial for resistance (or response) against M. brunnea or similar diseases in poplar.

General note 2 – selection of these 19 DIRs surely means that you are missing the response of other DIRs to phytohormones. Still this selection of the 19 DIRs can be understood if presented as deeper investigation into the gene expression regulation of these 19 DIRs. You might want to rephrase the text in lines 382 – 386 to explain it more clearly. This is just a comment and a final decision is up to you.

Line 404 – substitute “from” with “in”

Line 416 – delete “proteins” before reference 19

Line 430 – delete “relevant”

Line 431 – delete “P. trichocarpa”

Line 440-441 – substitute “repeated events” with “duplications”

Line 445 – what do you mean by “artificial selection pressure”? How could it be artificial if it happened so long ago?

Line 445-446 – delete the sentence “Gene replication also plays……. [58].”

Line 485 – change “plants” to “plant”

Line 488 – use “role of DIR genes in disease resistance in cotton…” instead of current form

Line 491 – use “The M. brunnea – responsive PeDIRs”, not “The PeDIRs involved in M. brunnea resistance”

Line 495 – insert “expression” between “gene” and “profiles”

Line 497 – use “occurred” instead of “increased”

Line 498 – insert “content” after “alcohol”

Line 507 – didn’t you analyze 19 genes, not 18?

Line 508 – 509 – the sentence “Various hormones…. disease and infection”. In what context are you using this sentence? If it is used in context of your experiments, I think you should delete this sentence. If you are using it as part of discussion, linked with the next sentences, again, these sentences are self-explanatory and more informative than this one. I think you should delete this sentence.

Line 511 – insert reference after “strongly”

Line 519 – use “pathways are not” instead of “pathway is not”

Line 528 – “provided an insight in the evolutionary history of the PtDIR genes”, not “provided evidence of…….”

Line 535 – delete “and their essential roles in poplar development and growth”

Line 537 – use “in response to”, not “under"

Author Response

Dear professor:

Thanks for your  valuable suggestions and comments to our last submitted manuscript. We modified the manuscript according to your comments.

Reviewer 3 Report

The authors report a genome-wide study of poplar derigent (DIR) proteins. They identified 40 DIR genes from Populus trichocarpa genome database and classified them into three groups, analysed cis-acting elements of promoter regions of each DIR gene. Furthermore, transcript level analysis in several organs and response on M. brunnea and various phytohormons was carried out.

The data and description in this Ms are clear and reliable and are informative for readers who are interested in DIR proteins. Therefore, this Ms is worth for publication in Forests.

I described minor revised points below.

L98  Pfam needs reference.

L146  What is “standard wood plant medium”?

L150  How big were 2 month-old seedlings? Were these seedling plants propagated by cutting?

Table 2 should not be a table, is Figure.

L358  Correct PeDIR23 to 16.

Figure 9 is too many data and small. The authors should pick up the data that want to stress and the other data remove to supplementary files.  

Author Response

Dear professor:

Thanks for the valuable suggestions and comments to our last submitted manuscript. We modified the manuscript according to your comments. Please see the attachment.

Round 2

Reviewer 1 Report

I have no concerns and suggest to accept the manuscript in the current form.